# Adaptive self-supervised learning of morphological landscape for leukocytes classification in peripheral blood smears

Zhuohe Liu
*Translational Molecular Pathology*
*The University of Texas MD Anderson Cancer Center*
Houston, USA
0000-0002-1818-646X

Simon P. Castillo
*Translational Molecular Pathology*
*The University of Texas MD Anderson Cancer Center*
Houston, USA
0000-0002-0606-2160

Xin Han
*Laboratory Medicine*

*The University of Texas MD Anderson Cancer Center*
Houston, USA
xinhan@mdanderson.org

Xiaoping Sun
*Laboratory Medicine*

*The University of Texas MD Anderson Cancer Center*
Houston, USA
xsun@mdanderson.org

Zhihong Hu
*Laboratory Medicine*

*The University of Texas MD Anderson Cancer Center*
Houston, USA
zhu6@mdanderson.org

Yinyin Yuan
*Translational Molecular Pathology*
*The University of Texas MD Anderson Cancer Center*
Houston, USA
0000-0002-8556-4707

*Abstract*—**Diagnosis of hematological disorders relies on cytomorphology and abundance of white blood cells (WBC) in peripheral blood smear (PBS). Hematology analyzers offer automated cell classification but cannot supersede manual review, especially in cancer patients where disease or treatment-induced morphological shifts necessitate frequent label corrections. To overcome both the rigid cell type definition and inefficient label usage, we developed a self-supervised model trained on image triplets with a lightweight EfficientNetV2-B0 encoder. With the learned morphological landscape, cell-type labels can be obtained using simple classifiers with tunable support sets, achieving a testing 9-way classification $F_1$ score of 96.2%. Moreover, our model readily generalizes to different label sets as demonstrated by predicting 11 morphological attributes. Active learning was used to further increase label efficiency without sacrificing performance. To enable wider adoption, the model was made accessible as a web application, HemoSight. We anticipate adaptive, accurate, and efficient self-supervised image classification to accelerate clinical workflow with morphological insight.**

*Keywords*—**image classification, self-supervised learning, active learning, hematology, peripheral blood smear**

## I. Introduction

Peripheral blood smear (PBS) captures white blood cells (WBC) with diverse cytomorphology from the dynamic process of hematopoiesis. PBS assay is an essential primary screening of many hematologic disorders because it reveals disease-characteristic cell type abundance and morphology [1]. In modern hematology laboratories, accurate classification of blood cells and fine-grained characterization of morphology are done by automated analyzers followed by hematopathologists' review [2].

However, the capability of hematology analyzers and the sequential nature of the workflow pose limitations. Automated analyzers have long been available and benchmarked to supplement direct microscopy review [3], but many platforms still require manual revision with much-anticipated accuracy improvement [4]. Especially in cancer hospitals, treatment and cancer-induced conditions lead to high rates of manual review and correction [5]. Because existing analyzers are designed to be disease-agnostic and case-independent, repeated corrections of misclassified cell type labels are needed even if patients share similar underlying disease contexts. Moreover, analyzers provide only a fixed number of cell type classes, limiting the ability to create customized cell groups based on morphological features [6], which currently have to be selected manually.

Recently, computer vision models have been developed to ease the bottleneck of manual PBS review, but studies prevalently used supervised methods that have limited generalization on cell types. For example, convolutional neural networks (CNN) were used to classify eight or fewer classes of blood cells, excluding abnormal leukocytes such as blasts [7], [8], [9]. In addition, lightweight models such as EfficientNet were adopted but only for 2-way classification of lymphocytes [10] or non-blast cells [11]. By design, supervised approaches require labeled data points across predefined classes, so pseudo-cell type classes like smudge or artifact need to be included to boost performance [12]. It can be argued that to exhaustively list all possible classes and prepare corresponding labeled examples to train a supervised model is impractical and inefficient, let alone classes defined by morphological features.

Self-supervised methods can be exploited to overcome the rigid class definition and the cost of label collection of supervised methods. Self-supervised learning is particularly advantageous when a vast amount of unlabeled data is available

and the number of relevant classes is unconstrained, such as in facial recognition [13]. With its growing adoption in medical imaging [14], [15], we hypothesize that self-supervised learning will increase the customization in problem-specific class definitions and the efficiency of labeled data usage (e.g. [16]). Furthermore, because labels are used only after lengthy pretext training, self-supervised models can benefit from accumulating labels and label corrections, to incrementally improve performance [17]. Incidentally, this added flexibility also addresses limitations of semi-supervised approaches by improving data utilization and offering task agnostic learning [18].

Here, we present a label-efficient and adaptive self-supervised model to classify digitized WBC image crops of leukocytes from PBS analyzers. The model establishes a morphological landscape using similarity learned by image triplets, whose features are extracted by fast EfficientNetV2 encoder. We demonstrate that the model is adaptable to different proportions of labeled data, and different label domains (cell type vs. morphological attributes), with sufficient performance. By incorporating active learning strategies, label efficiency can be further improved without compromising performance. We also evaluated the model's explainability using visualizations. Recognizing its potential significance in clinical applications and training, we encapsulated our model in a web application, HemoSight, that offers interactive exploration of morphological space and real-time model inferences.

## II. METHODS

### A. Data and Annotation Collection

We collected blood samples from 44 randomly selected patients admitted to the University of Texas MD Anderson Cancer Center in 2023. PBS slides were prepared by Wright's staining and then scanned by Sysmex DI-60 automated digital cell morphology system. Using CellaVision software, about 115 cells were sampled from each slide to obtain image crops of about 360 × 360 pixels at 100× of magnification, each crop containing one white blood cell at the center. The labels of image crops were reviewed by three hematopathologists iteratively, based on the initial classification from CellaVision. We kept the class "blast" which was defined by the merged population of blast cells (e.g. myeloblasts, lymphoblasts, monoblasts) and blast equivalent cells (i.e. promonocyte). For the rest of the WBC classes of interest, we chose to use public dataset [7], [19], which included 17,902 images spanning eight classes: neutrophil, eosinophil, basophil, lymphocyte, monocyte, immature granulocyte ("ig"), erythroblast and platelet. The "ig" class contains promyelocytes, myelocytes, and metamyelocytes. In total, we appended 5,952 images to the public data to form a complete dataset of 23,044 images across nine classes with the following distribution: basophil (5.3%), blast (11.9%), eosinophil (13.8%), erythroblast (6.7%), ig (12.6%), lymphocyte (8.7%), monocyte (9.3%), neutrophil (21.4%), and platelet (10.2%).

From this dataset, 10% of the images (2,305 cells) were randomly selected with stratified sampling across cell types to form the holdout test set. The rest of the data was further split by 5-fold stratified sampling to form the training set (~16,591) and the validation set (~4,148).

This study received ethical approval from the Institutional Review Board of the University of Texas MD Anderson Cancer Center.

### B. Self-supervised Pretext Training

Our self-supervised model consists of three encoder heads with shared weights followed by a similarity layer, we implemented the model using TensorFlow 2.10.1 and Python 3.8.10.

We opted EfficientNetV2-B0 as the feature encoder due to its small size and thus requires less computation resources and time [20]. Incoming images were color-normalized by the gray world algorithm [21]. Image augmentation included random horizontal and vertical flipping, rotation (±90°), zoom (±0.1×), horizontal and vertical translation (±20 pixels), and color jitter (brightness, contrast, saturation, and hue, with 0.2 magnitude). Finally, images were center cropped to 224 × 224 pixels before feeding to the encoder. Note that pixel intensity rescaling from 0-255 to 0-1 was handled by the TensorFlow model internally.

The output feature vectors from the top activation layer of the encoder were first passed through a global average pooling layer, and then a dropout layer with a rate of 0.2. Finally, the vectors were $L_2$-normalized by a lambda layer before computing the loss.

The pretext similarity-based training requires anchor-positive-negative triplets, where anchor and positive images are generated from augmentations of the same file, and the negative image is taken from a different file. In practice, when triplet semi-hard loss [13] from the TensorFlow Addons library (version 0.20) was used, the image generator supplied two augmented versions of all images in a batch to enable efficient online triplet mining.

To train the network, we initialized the model using weights pre-trained on ImageNet [22]. The model was trained for all layers except batch normalization layers were frozen [23]. Semi-hard loss with a margin of 0.5 and $L_2$ distance metric was used. The model was trained for 30 epochs at a batch size of 100. Adam optimizer was used with a learning rate of $10^{-5}$, and default parameters of $\beta_1 = 0.9$, $\beta_2 = 0.999$, $\epsilon = 10^{-7}$. We manually explored the hyperparameter space including optimizer, learning rate, learning rate schedule, loss margin.

We containerized the script to a Docker container and deployed it on the Kubernetes cluster at our institution. For each job, one NVIDIA A100 GPU of a node was used with 40 GB of graphics memory. The script is available on GitHub: https://github.com/MXGHarryLiu/HemoSight.

### C. Supervised Fine-tuning and Label Set Generalization

To obtain classification labels, a support set was created from the training set using the embeddings from the pretext-trained encoder and the corresponding hematopathologist-provided labels. A linear support vector machine (SVM) classifier [24] was used to predict categorical labels from embeddings. The SVM used one vs. rest decision function shape for multi-class classification and with the input standardized and probability estimates enabled for the output. For the following investigations, the learned embedding space was kept the same while the SVM was retrained for different tasks.

To test the impact of labeled data proportion on the performance, we random-sampled an increasing number of labeled data stratified for each cell type to be included in the support set.

To generalize the model, we applied the model to a subset of data with 11 morphological attributes obtained from the literature [25]. Because not all images have morphological annotations, we kept the same data split while using only the intersection of data points with available labels. This yielded ~7,379 training images and ~1,826 validation ones. We repeated the classification for each morphological attribute with the same classifier settings.

To account for class imbalance, we report macro-averaged scores.

### D. Supervised Baseline Model

To serve as a baseline comparison to our self-supervised model, we implemented a simple supervised classifier with the same EfficientNetV2-B0 encoder. We kept a similar architecture to the self-supervised model except one encoder head was used and a dense layer with Softmax activation was added after the dropout layer to generate class probability and labels for 9-way classification. As commonly used in transfer learning, we performed two-stage training using the Adam optimizer, a batch size of 32, and the sparse categorical cross-entropy loss. First, the model was trained at a learning rate of 0.001 with all layers except the top layer frozen for 10 epochs. Then, we unfroze all layers except the batch normalization layers and trained the model for an additional 20 epochs at a lowered learning rate of $10^{-5}$ to fine-tune the weights. To prevent overfitting and ensure convergence, when the training data size is at or below 450 images, the second stage is skipped, and the model is trained for 30 epochs with only the top layer unfrozen.

### E. Active Learning

To improve label efficiency, we simulated active learning strategies implemented by the modAL package (version 0.4.2.1) [26] on our dataset with cell-type labels. We initialized the model with 90 labels, randomly sampled and stratified across cell types. Then, inferred labels and their probabilities from SVM were used to determine the next batch of 90 samples to be labeled, using two sampling strategies: (a) uncertainty sampling, which selected samples with the highest uncertainty, i.e. lowest max probability across all classes, and (b) random sampling, which was used as a baseline. Note that this cumulative random sampling is different from random resampling of the training set with increasing sample size as we did in the above section. For each iteration, performance scores were logged, and the process was repeated until all training samples were used.

To compare sampling strategies, we obtained a learning curve, i.e. performance ($P$) vs. sample size ($n$), for each class, and fitted it to an inverse power-law curve (1), where $P_{max}$ is the asymptote performance, $r$ is the learning rate, $k$ is the scaling factor, and $n_0$ is the offset.

$$P(n) = P_{max} - \frac{k}{(n-n_0)^r} \qquad (1)$$

### F. Explainability

To visualize the learned embedding space of the self-supervised model, we employed openTSNE (version 1.0.1) [27], an implementation of the t-distributed stochastic neighbor embedding (t-SNE) to reduce the dimensionality from 1280 of the output feature vectors to two. Euclidean distance was used, and the embedding was initialized using top principal components. 1000 iterations were used at a perplexity of 30.

Gradient-weighted class activation mapping (GradCAM) is another technique for visualizing model decisions. However, it is commonly applied to supervised learning thus "class activation". Inspired by a few studies that extended GradCAM to the Siamese network [28], we adapted the GradCAM++ algorithm [29] to our self-supervised triplet network. Output (7×7×1280) from the top activation layer was exported to compute the activation map (7×7). Because online triplet mining was used, informative triplets need to be inferred by images with the top matrix sum of activation maps for a given batch with a non-zero loss.

### G. Web Application Packaging and Deployment

To ease access to the model, we developed a full-stack web application consisting of four Docker containers. First, a ReactJS container served the frontend web pages, which adopt styles from the Bootstrap framework. RESTful APIs were defined to send and receive data between the frontend and the backend container, which uses FastAPI (version 0.105) in Python. To improve server responsiveness, a third container running the TensorFlow models in Python was dedicated to model inference. This worker container, which also uses FastAPI, subscribed to the change stream of a database container (MongoDB) to receive user requests of incoming image classification jobs and push results back to the database. With this design, theoretically, multiple instances of this worker container can be deployed to speed up model inference. The results were visualized as interactive Vega charts in the web browser [30].

### III. RESULTS

### A. Adaptive Leukocytes Classification Based on Self-supervised Triplet Model

As labeled data are typically obtained at a higher cost than unlabeled data, we first evaluated model performance with varied sizes of the support set, thus the proportion of labeled data in the training set. For each class of 9 cell types, we selected data points to be used for SVM prediction using stratified random sampling. We saw the self-supervised model can tolerate few labeled data. For example, when 5.4% of labeled data were used, the $F_1$ score only dropped 3.86% compared with that of 96.32% if all labels were used (Fig. 1). This demonstrates that our model can readily accept an expanding support set upon the availability of new labels.

In addition, the self-supervised model surpasses supervised baseline models for label counts below 1,800, and when all labels were used, the $F_1$ score of 96.32±0.25% is close to 97.47±0.22% of supervised models (Fig. 1). When the hold-out set was evaluated, the model achieved an $F_1$ score of 96.20±0.09% across 5 repeats. This shows that our model

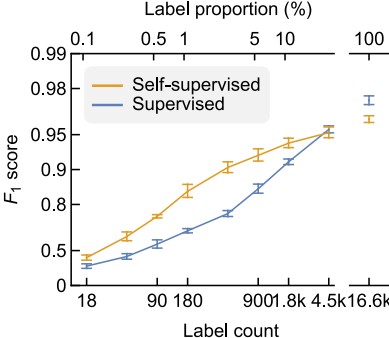

Fig. 1. Cell type prediction performance vs. proportion of labeled data. Values are mean across 5 folds of cross-validation, and error bars are std. y-axis is scaled by $-\log(1-x)$.

achieved consistent and sufficient performance.

Then, we sought to investigate how the model can adapt to different attribute sets. Replacing cell-type labels with categorical labels of various WBC morphological attributes, we repeated the performance evaluation by retraining the classifier on the same embedding space. Our model achieved good performance, with an averaged mean $F_1$ of 85.78±0.34% across all 11 morphological attributes (Fig. 2), which is comparable to published results from supervised models (91.20±0.06%) [25].

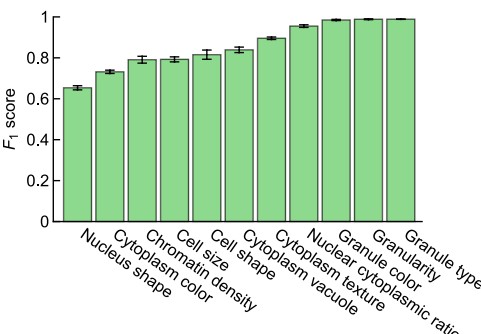

Fig. 2. Self-supervised model's performance across 11 morphological attributes. Values are mean across 5 folds of cross-validation, and error bars are std.

Next, to further improve label efficiency, we tested the self-supervised model's performance with active learning strategies. Using uncertainty sampling, only 2250 labels (13.3% of total, after 25 iterations) were needed to have the performance reaching that of using all labels (Fig. 3).

In addition, different cell types show different learning rates response to the same sampling strategy, with platelet raising the fastest, and monocyte the slowest (Fig. 4a). An increased learning rate might be caused by decreased intraclass variation or increased interclass separation. The learning rate improvement can be seen across 8 of 9 cell types and is significant compared with random sampling ($p = 0.006$ one sided Wilcoxon signed-rank test) (Fig. 4b). This indicates that active learning strategies can drastically cut down labeled data usage by focusing on only informative data points, and labeling can be further guided by cell type difference to account for different intraclass variations.

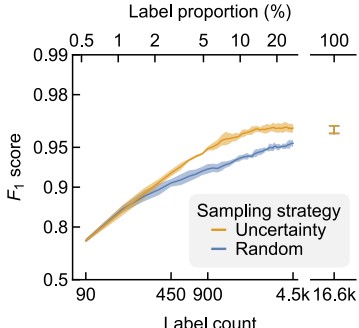

Fig. 3. Learning curves of active learning sampling strategies. Mean macro-averaged $F_1$ score vs. increasing support set across 5 folds of cross-validation. Shaded areas denote std. y-axis is scaled by $-\log(1-x)$.

(a)

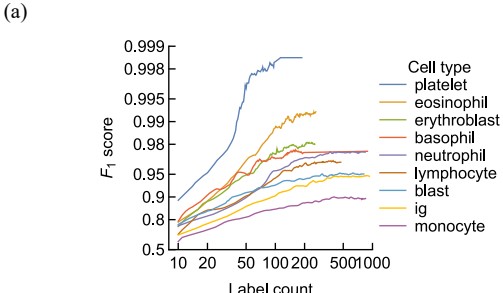

(b)

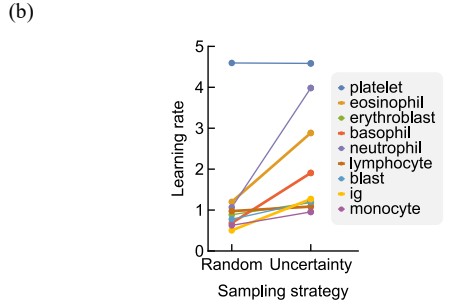

Fig. 4. Learning curves vs. active learning sampling strategies. (a) Learning curves of individual cell types during uncertainty sampling. Mean values across 5 folds of cross-validation are shown. y-axis is scaled by $-\log(1-x)$. (b) Learning rates obtained from mean learning curves of individual cell types from different active learning sampling strategies.

Finally, to explain decisions made by the model, we first visualize the embedding space using t-SNE. Clear clustering by cell type can be seen in the t-SNE map (Fig. 5a). Interestingly, we can also see clusters of eosinophils and neutrophils showing polarized distribution of cell size (Fig. 5b), indicating the embedding space encodes similarity defined by both cell types and morphological attributes. In addition, to visualize model spatial attention, we computed class activation maps which allow us to confirm the model's decision for similarity is mainly drawn from the center region, instead of the background or surrounding red blood cells (Fig. 6). We also noticed that strong activation often came from image pairs of the same underlying cell type, indicating the effect of semi-hard loss filtering and effectiveness of self-supervised training.

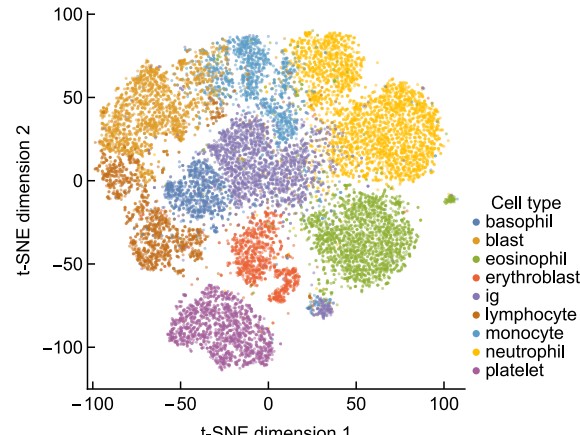

(a)

(b)

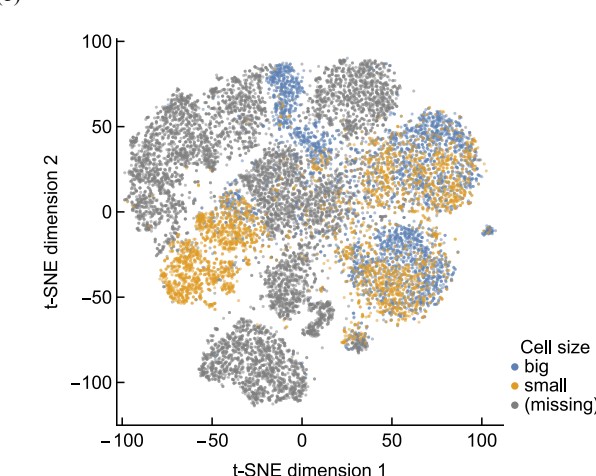

Fig. 5. Visualization of morphological landscape by t-SNE. (a) Scatter plots showing embedding space dimension reduced by t-SNE of a representative model trained using one of cross-validation data splits. Data points colored by cell type. (b) Same as (a) but colored by cell size morphological attributes.

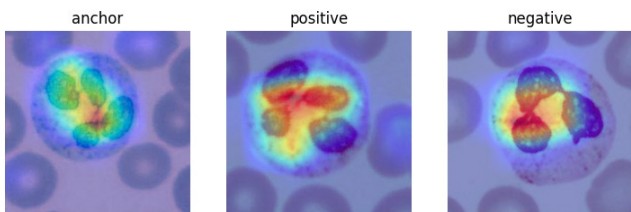

anchor          positive          negative

Fig. 6. Class activation maps explain model decision spatially. Class activation maps of two neutrophils superimposed on corresponding input images. Anchor and positive are obtained from augmentations of the same image, while negative is obtained from a different image. Red indicates strong activation, while blue indicates weak activation.

## B. Web Application for AI-assisted Peripheral Blood Smear Review

To enable wider adoption of our machine learning model in future AI-assisted workflow of peripheral blood smear review, we deployed our trained model to a web application, named HemoSight. HemoSight supports batch uploading of query images for cell type inference (Fig. 7a). Despite no parallel

(a)

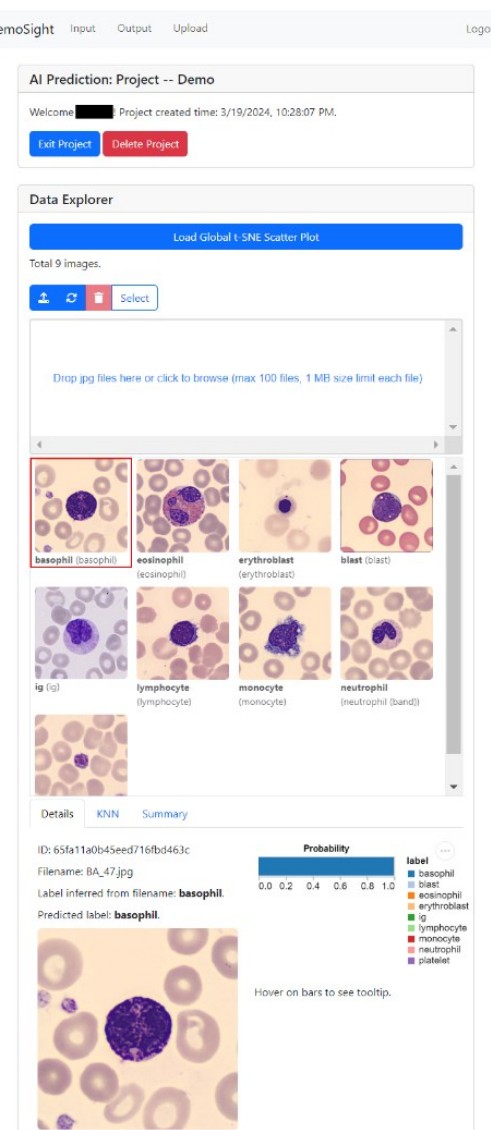

(b)

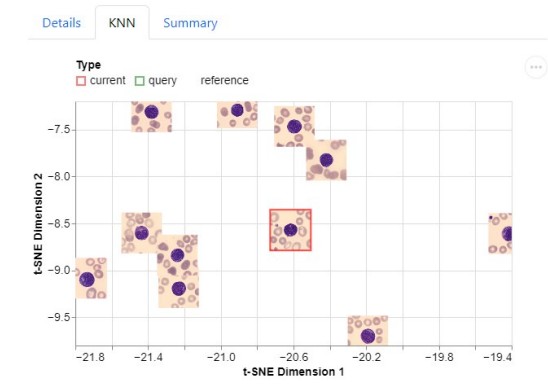

Fig. 7. Features of the HemoSight web application. (a) HemoSight enables users to upload query images and perform cell type inference. (b) Given one query image, its nearest neighbors in the embedding space are visualized.

processing being used, the server was able to process incoming data at a speed of tens of images per second. For a given query image, probabilities of predicted labels are shown in the "details" tab to provide confidence of estimation (Fig. 7a).

Moreover, using openTSNE [27], HemoSight can remap query images onto the learned embedding space, enabling visualization of neighboring similar images to be used for manual reference (Fig. 7b).

## IV. Discussion

In this paper, we demonstrated a label-efficient model for classifying WBC crops from peripheral blood smear into flexibly defined label sets such as cell type or morphological attributes. We showed that such a self-supervised model can tolerate label-scarce datasets with minimal tradeoffs in performance. The adoption of active learning sampling strategies further lowered the threshold for label proportion to achieve target performance. We visually examined the explainability of the model before deploying the model to a proof-of-concept web application.

Our work is not without caveats or limitations, which present opportunities for future studies. On model assumptions, we defined similar images using augmentation of the entire image, and only one embedding is obtained per image to compute the distance metric. This means the model will have limited discernment of local features. In practice, subcellular morphologies such as Auer rods and hairy cell membrane are important hallmarks for certain diseases [6]. To detect these local features, examples from different cell types, and thus different dominant global features, are necessary, as in the case of vacuole, which may be the source of label inefficiency. We could explore alternative augmentation strategies or model frameworks to capture local similarity.

Moreover, by supplementing the public dataset with our internal dataset, we hope to expose the model to systematic variations. However, we didn't quantify batch differences and the impact of data normalization. Given we can see some intra-class separations of data from different sources (Fig. 5), the robustness of the model can be potentially boosted by including training data from more institutions.

On model implementation and evaluation, more complicated classifiers, such as tree-based ensemble methods, can be used instead of SVM. However, choices of classifier may influence pretext hyperparameter tuning and most downstream evaluations, and computation cost as well as overfitting from additional hyperparameters need to be monitored. In addition, other active learning sampling strategies can be explored, as well as the impact of initialization [31] or query batch size.

We created the web application not only to show the benefit of fast and efficient machine learning models that could enable quick model inference but also to demonstrate the potential of integrating real-time label correction and post-training class definition to clinical workflows, when self-supervised models are paired with active learning. Realizing the latter in a fully-fledged web application, however, is beyond the scope of this study.

Our implementation of self-supervised learning should be domain agnostic, and we anticipate that the same approach can be applied to other modalities such as cytology samples of bone marrow[32], [33] or imaging flow cytometry [34]. However, additional preprocessing steps such as cell segmentation are necessary for densely populated images, and transfer learning may be required for cells in suspension. In addition, we could establish patient cohorts across diagnostics to study the effect of disease on morphology [35].

Our study lays the foundation for AI-assisted peripheral blood smear review where machine learning models iteratively improve performance using expended human annotations on top of knowledge of image similarity mined from the vast amount of unlabeled data. We expect self-supervised models to shorten the turnaround time of model updates, leading to a better understanding of hematological disorders and their impact on morphology.

## Acknowledgment

We thank Xiaoxi Pan for her expertise and discussion in model training. This work is supported by seed funding from the University of Texas MD Anderson Cancer Center.

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
