# OpenReview forum: "Adaptive self-supervised learning of morphological landscape for leukocytes classification in peripheral blood smears"
_IEEE.org/EMBS/BHI/2024/Conference — IEEE BHI'24_

### Official Review · Reviewer_6Z8a · 2024-08-09
**User-interface tool to automate classification of blood cells**

**Overall Rating:** 7
**Confidence:** 4

**Other Quality Metrics:**

(a) Clarity of writing; the paper is well-written and well-structured. The number of figures might be too much. This could be addressed by merging some figures such as 1 and 3 and taking some to an appendix such as 2.

 (b) Clinical Significance; the paper is directly addressing a clinical problem which is automating blood cell classification in terms of size and types.

 (c) Methodological Novelty; The classification method specially the developed software seems novel.

(d) Experiments and Results; the used dataset and the data analysis seems convincing in showing the effectiveness of the proposed classification.

I truly enjoyed reading this well-prepared paper.

**Questions For The Authors:**

On top left of page 4, the two line descriptions should belong to the caption of Fig. 1 with the light blue color. If the style is only to have the first sentence in blue, then please keep the whole caption in a same page.

It is not clear why there is a cut between 10 and 100% in Fig. 1 owing to the fact that it is in logarithmic scale. Is it due to the time consumption of processing a large range of labels?

Please remove ‘(a)’ in the caption of Fig. 3.

Would it be possible to plot the data between 20% and 100% proportion in Fig. 3?

In the final version, please try to fill the empty gap on page 4.

Is HemoSight open source to use?

Touching upon application of your white blood cell classification model in other modalities, please discuss how you envision the performance of it when it comes to images of blood samples with a certain flow rate and/or located in an in-vitro micro vessel such as those in Fig. 6 or Fig 7 of https://doi.org/10.1038/s41598-021-97008-w

Please explain why the learning rate of platelets in terms of sampling strategy in Fig. 4(b) remains the same.

**Strengths:**

Presentation of a label-efficient and adaptive self-supervised model for the classification of white blood cells images. Through image overlay on the red blood cells, it has been shown that the proposed self-supervised learning model is effective in cell classification.

A user-interface-based peripheral blood smear review package called HemoSight is developed for accelerated and convenient classification purposes.

**Summary Of The Paper:**

In this paper a self-supervised model is developed that is trained on image triplets an encoder called lightweight EfficientNetV2-B0.

**Weaknesses:**

The findings/major contribution of this paper is not mentioned in the abstract.

Specific application of this research is not mentioned in the Key Words.

---

### Official Review · Reviewer_Ek2D · 2024-08-16
**Evaluation on the model trained with supervised and self-supervised approach to classify white blood cells from blood smear images.**

**Overall Rating:** 7
**Confidence:** 5

**Other Quality Metrics:**

(a) Clarity of writing: good
(b) Clinical Significance: good
(c) Methodological Novelty: great
(d) Experiments and Results: good

**Questions For The Authors:**

1. The combined dataset was created from 5952 cell images from public domains and the rest images from in-house dataset yielding a total 23k images. I suggest that authors could use their in-house dataset for model development and used the public data to evaluate the model. In this approach, the public dataset is considered as indecent testing. This approach is to prevent the institutional data bias and demonstrate the model could generalized well at least in 2 public datasets.
    2. The results shown the model performance  between supervised vs self-supervised approach, however, a benchmark between these models performance with common tradition machine learning models such as XGBoost and LightGBM.
    3. The class-activation map demonstrated the model decision shown on neutrophil class only, authors could display the rest 8 cell types for a comprehensive understanding of model activation.
    4. The self-supervised model trained with 11 morphological attributes is also interesting. I wish to see the GRAD-CAM on these morphological based groups to see whether the model could find the hot-spot for its decision.

**Strengths:**

The current work developed model using standard practice in the machine learning field with k-fold cross validation and hold-out set for testing. The proposed methods could be generalized in any imaging-based datasets.

**Summary Of The Paper:**

The current manuscript proposed a deep learning approach to predict the white blood cells from peripheral blood smear samples. The model was trained and tested a mixed dataset combining in-house and public datasets. The model performance yield a good accuracy on the hold-out set. Authors also shown the class-activation map to support the model decision interpretation.  Overall, the current work is scientifically sounds, the findings also support the authors claims.

**Weaknesses:**

I have commented the weakness to the authors. In brief, the dataset was mixed from in-house dataset with public datasets for both model development and testing. This approach makes the readers considering that the model was trained/tested/evaluated in different portions of the dataset with k-fold. instead of merging these datasets, author could keep at least 1 public dataset to serve as the external independent dataset, which could definitely boost the confidence on the model performance.

---

### Official Review · Reviewer_EgL9 · 2024-08-18
**Adaptive Self-Supervised Learning of Morphological Landscape for Leukocytes Classification in Peripheral Blood Smears**

**Overall Rating:** 7
**Confidence:** 5

**Other Quality Metrics:**

(a) Clarity of Writing: Great
(b) Clinical Significance: Great
(c) Methodological Novelty: Excellent
(d) Experiments and Results: Great

**Questions For The Authors:**

1. The authors use t-SNE to visualize one of the 5-cross validation results, providing a clear interpretation of what the model has learned. Did the other four cross-validation folds demonstrate similar patterns?
2. The authors mention that current analyzers provide only a fixed number of cell type classes, limiting the ability to create customized cell groups. Based on the authors’ experience, what is the average proportion of these cell types typically identified in daily scenarios at a cancer hospital? This information is crucial for determining the appropriate cutoff proportion in their benchmark test when comparing with other supervised models.

**Strengths:**

1. The self-supervised approach reduces dependency on large labeled datasets, demonstrating superior performance even with limited labeled data.
2. Active learning strategies enhance label efficiency, enabling the model to achieve high accuracy with fewer labeled examples.
3. Deploying the model as a web application increases its accessibility and practical relevance, potentially streamlining clinical workflows.
4. The study provides a clear evaluation of model explainability through t-SNE visualization and class activation maps, ensuring transparency in the model’s decision-making process.

**Summary Of The Paper:**

This paper presents a self-supervised learning model developed to classify leukocytes in peripheral blood smears by analyzing their morphological features. The model, based on EfficientNetV2-B0, employs image triplets during training to effectively capture the similarities and differences among various leukocyte types. The study demonstrates that the model performs comparably to traditional supervised models, with particular strength in scenarios where labeled data is scarce. Additionally, the model integrates active learning strategies to improve label efficiency without sacrificing classification accuracy. It has been implemented in a web application, offering a tool for clinical workflows that provides adaptive, accurate, and efficient leukocyte classification.

**Weaknesses:**

1. The benchmark on alternative model architectures, including the potential benefits of classifiers beyond SVM, is limited. Expanding these tests could offer a more comprehensive view of possible improvements.
2. The model underperforms relative to supervised learning when all labels are used, making it challenging to apply clinically as a replacement for current algorithms.
3. The manuscript lacks a detailed discussion on the varying learning rates of different cell types in response to the same sampling strategy, which is crucial for understanding the model’s behavior under various conditions.

---

### Decision · Program_Chairs · 2024-09-23

Accept